# Phosphatidic acid modulation of Kv channel voltage sensor function

**Richard K Hite, Joel A Butterwick, Roderick MacKinnon***

Laboratory of Molecular Neurobiology and Biophysics, Howard Hughes Medical Institute, Rockefeller University, New York, United States

**Abstract** Membrane phospholipids can function as potent regulators of ion channel function. This study uncovers and investigates the effect of phosphatidic acid on Kv channel gating. Using the method of reconstitution into planar lipid bilayers, in which protein and lipid components are defined and controlled, we characterize two effects of phosphatidic acid. The first is a non-specific electrostatic influence on activation mediated by electric charge density on the extracellular and intracellular membrane surfaces. The second is specific to the presence of a primary phosphate group, acts only through the intracellular membrane leaflet and depends on the presence of a particular arginine residue in the voltage sensor. Intracellular phosphatidic acid accounts for a nearly 50 mV shift in the midpoint of the activation curve in a direction consistent with stabilization of the voltage sensor's closed conformation. These findings support a novel mechanism of voltage sensor regulation by the signaling lipid phosphatidic acid.

**\*For correspondence:** mackinn@rockefeller.edu

**Competing interests:** The authors declare that no competing interests exist.

## Introduction

Voltage-gated potassium (Kv) channels shape and terminate action potentials. While membrane voltage is the fundamental stimulus for Kv channel gating, other stimuli such as protein phosphorylation (*Vacher and Trimmer, 2011*), intracellular $Ca^{2+}$ (*Gamper et al., 2005*) and accessory proteins also regulate various Kv channels.

In retrospect, given the lipid complexity of cell membranes, it is not surprising to learn that specific lipid molecules are among the regulators of membrane proteins generally and ion channels specifically (*Hilgemann et al., 2001*; *Dart, 2010*). Even the structurally simple bacterial $K^+$ channel KcsA requires anionic phospholipids in order to open (*Heginbotham et al., 1998*), while the more complex eukaryotic inward rectifier $K^+$ channels are so dependent on the lipid $PIP_2$ (see abbreviations section for lipid and detergent definitions) that they might have been called the $PIP_2$-regulated $K^+$ channels (*Huang et al., 1998*; *Suh and Hille, 2005*; *Hansen et al., 2011*; *Whorton and MacKinnon, 2011*). These are just two examples from a growing list of ion channels whose function depends on the presence of specific lipid molecules. In the latter example, since $PIP_2$ levels vary as a function of the physiological state of a cell, $PIP_2$ is known as a 'signaling lipid' because it triggers or signals the action of molecules to which it binds, such as inward rectifier $K^+$ channels.

The function of certain Kv channels is also strongly influenced by the membrane's lipid composition. For example, the archeal KvAP channel requires phospholipids in order to open; they remain completely inactive in non-phospholipid membranes (*Schmidt et al., 2006*). Eukaryotic Kv7 channels, also known as M current channels, depend on the presence of $PIP_2$ in the membrane (*Falkenburger et al., 2010*; *Kruse et al., 2012*; *Rodriguez-Menchaca et al., 2012*; *Telezhkin et al., 2012*). Stimulation of Gq-coupled G protein coupled receptors activates phospholipase C, which depletes $PIP_2$ from the membrane's inner leaflet and closes Kv7 channels (*Dickson et al., 2013*). Thus, $PIP_2$ functions as a signaling lipid in regulating Kv7 channel activity, just as for inward rectifier $K^+$ channels.

**eLife digest** The electrical signals that carry information through the nervous system rely on positively charged potassium ions moving in and out of neurons. These ions move through proteins called voltage-gated potassium channels that are embedded in the plasma membrane that surrounds the neurons. The potassium channels contain pores that can be opened and closed to control the movement of the potassium ions.

The main factor that controls the opening and closing of these channels—a process known as 'gating'—is the voltage across the membrane. However, the channels can also be controlled by proteins, or by other molecules.

The plasma membrane is made of several different types of molecules called phospholipids. Some of these phospholipids are known to be involved in gating potassium channels, but the roles of other phospholipids remain unclear.

To investigate the role of a phospholipid called phosphatidic acid, Hite et al. placed potassium ion channels in artificial plasma membranes. These experiments revealed that phosphatidic acid alters the gating of potassium ion channels in two ways. The first way is generic: the negative charge in phosphatidic acid shifts the membrane voltage. The second way is specific to phosphatidic acid: the end of the molecule with the negative charge interacts with the part of the potassium channel that senses changes in voltage to keep the pore closed. The next challenge is to understand how neurons shift their phosphatidic acid levels to regulate their electrical activity.

In this study we report our findings from a systematic analysis of Kv1 channel dependence on membrane lipid composition. Channels were reconstituted into planar bilayer membranes, which allow complete control of lipid composition (*Miller, 1986*). We identified one lipid, phosphatidic acid (PA), which uniquely affects channel gating. We next studied the mole fraction dependence, membrane sidedness, and chemical characteristics of PA necessary to mediate its effect. We further show that PA similarly alters gating in the distantly related Kv channel KvAP. Experiments with mutant KvAP channels point to a mechanism whereby the primary phosphate group on PA stabilizes voltage sensor arginine residues in the closed conformation. Because PA is a naturally occurring 'signaling lipid' in the inner leaflet of cell membranes, we think we have likely uncovered a new and biologically relevant mode of Kv channel regulation.

## Results

### Phospatidic acid regulates Kv channel gating

*Figure 1* introduces the fundamental observation that this study seeks to understand: phosphatidic acid is an outlier among tested lipids in its ability to influence voltage-dependent gating of a $K^+$ channel.

The $K^+$ channel under study is a mutant of the rat Kv1.2 channel in which the helix-turn-helix segment termed the voltage sensor paddle was replaced by the corresponding segment from Kv2.1, a closely related $K^+$ channel (*Long et al., 2007*). This 'paddle chimera' mutant is more stable biochemically but otherwise is functionally very similar to wild type (*Tao and MacKinnon, 2008*). The α (conduction pore and voltage sensor-forming) subunit was expressed and purified with its β subunit, an aldo-keto reductase-like domain, attached to the cytoplasmic surface (*Gulbis et al., 1999*). We refer to the α-β complex of the mutant channel simply as the Kv channel.

When reconstituted into planar lipid bilayers the Kv channel opens upon membrane depolarization from a negative (inside relative to outside) holding voltage (*Figure 1A*). *Figure 1B* graphs tail currents (normalized to maximal current), which are measured shortly after stepping negative from the depolarization voltage, as a function of the depolarization voltage. This voltage-dependent 'activation curve' shows that channels begin to open around −100 mV and reach near maximal activation by −40 mV, with a half activation voltage ($V_{mid}$) of approximately −70 mV. These currents were recorded in DPhPC bilayers, which were chosen as the 'baseline' lipid in this study because they form exceptionally stable bilayers. *Figure 1D* shows the influence of mixing different lipids at a mole fraction of 0.25 into the DPhPC lipid (*Figure 1—figure supplement 1*). In all but one case the activation curves are similar, with $V_{mid}$ around −70 mV. POPA is unique: it produces an activation curve that is less steep and has a $V_{mid}$ around −40 mV.

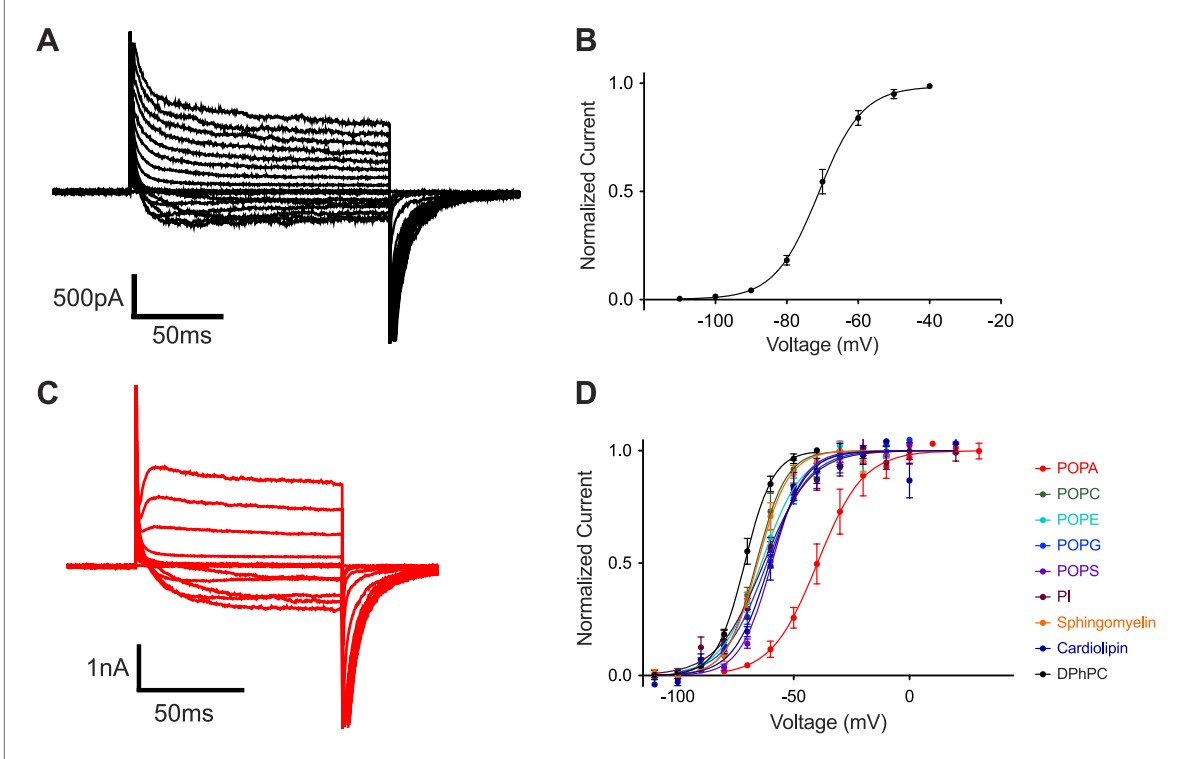

**Figure 1**. POPA modifies Kv channel gating. (**A**) Representative family of currents recorded from Kv channels in DPhPC bilayers. Voltage is stepped from a holding voltage of −110 mV to increasingly more positive depolarization voltages (−110 mV to +80 mV; ΔV = 10 mV) and then returned to the holding voltage of −110 mV. (**B**) Normalized tail currents (mean ± SEM) from current families recorded from Kv channels in DPhPC bilayers are fit with a Boltzmann function with half activation voltage $V_{mid}$ = −71 ± 1 mV, Z = 4.2, N = 8. (**C**) Representative family of currents recorded from Kv channels in DPhPC:POPA (3:1) bilayers from a holding voltage of −80 mV to increasingly more positive depolarization voltages (−80 mV to +40 mV; ΔV = 10 mV) and then returned to the holding voltage of −80 mV. (**D**) Normalized tail currents (mean ± SEM) from current families recorded from Kv channels in different lipid mixtures are fit with Boltzmann functions (DPhPC:POPA (3:1) $V_{mid}$ = −40 ± 2 mV, Z = 2.6, N = 7; DPhPC:POPC (3:1) $V_{mid}$ = −66 ± 1 mV, Z = 4.0, N = 6; DPhPC:POPE (3:1) $V_{mid}$ = −64 ± 1 mV, Z = 3.0, N = 7; DPhPC:POPG (3:1) $V_{mid}$ = −62 ± 1 mV, Z = 2.9, N = 9; DPhPC:POPS (3:1) $V_{mid}$ = −59 ± 1 mV, Z = 3.8, N = 6; DPhPC:PI (3:1) $V_{mid}$ = −63 ± 1 mV, Z = 2.6, N = 7; DPhPC:Sphingomyelin (3:1) $V_{mid}$ = −66 ± 1 mV, Z = 3.9, N = 6; DPhPC:Cardiolipin (3:1) $V_{mid}$ = −60 ± 1 mV, Z = 3.4, N = 6; DPhPC $V_{mid}$ = −71 ± 1 mV, Z = 4.2, N = 8).

The following figure supplement is available for figure 1:

**Figure supplement 1**. Representative families of currents recorded from Kv channels.

At a mole fraction of 0.25, POPA induced a rightward shift of the activation curve (*Figure 1D*) that is associated with slowed activation kinetics (*Figure 1A,C*). *Figure 2* shows to what extent channel gating is altered as the mole fraction of POPA is varied (*Figure 2A,B*, *Figure 2—figure supplement 1*). At a mole fraction of 0.05 the effect of POPA on $V_{mid}$ is already substantial, and by 0.1 it is nearly complete. Thus, POPA influences gating according to an approximately saturating function with a steep dependence in the low (less than 0.1) mole fraction range. The functional relationship is similar whether POPA is added to DPhPC or POPE membranes, although, due to the intrinsic instability of pure POPE bilayers, gating was not assessable at the origin in POPE (*Figure 2B*).

The data graphed in *Figure 3* show how chemical variation within the lipid head-group or acyl chain affects $V_{mid}$ (*Figure 3—figure supplement 1*). Here, as in *Figure 2*, the lipid under study was added to DPhPC and $V_{mid}$ was graphed as a function of the added lipid mole fraction. In *Figure 3A* five lipids with identical or similar acyl chains but different head-groups are compared (*Figure 3A,E*, *Figure 3—figure supplement 1*). Only POPA has a large effect. POPG, POPS and PI, similar to POPA, have a net −1 charged head-group. Thus, the large gating effect of POPA on $V_{mid}$ is not attributable to the −1 charge of the head-group.

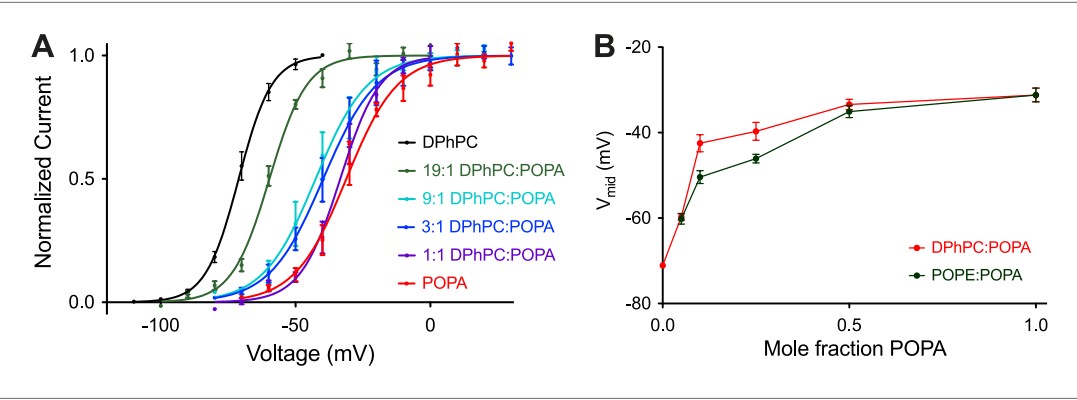

**Figure 2**. Concentration dependence of Kv channel activation by POPA. (**A**) Normalized tail currents (mean ± SEM) from current families recorded from Kv channels in DPhPC:POPA mixtures are fit with Boltzmann functions (DPhPC $V_{mid}$ = −71 ± 1 mV, Z = 4.2, N = 8; DPhPC:POPA (19:1) $V_{mid}$ = −60 ± 1 mV, Z = 3.7, N = 8; DPhPC:POPA (9:1) $V_{mid}$ = −43 ± 2 mV, Z = 2.7, N = 5; DPhPC:POPA (3:1) $V_{mid}$ = −40 ± 2 mV, Z = 2.6, N = 7; DPhPC:POPA (1:1) $V_{mid}$ = −33 ± 1 mV, Z = 3.5, N = 6; POPA $V_{mid}$ = −31 ± 2 mV, Z = 2.7, N = 6). (**B**) Plot of $V_{mid}$ determined from fit of tail currents to the Boltzmann equation vs mole fraction of POPA for Kv channels in bilayers containing DPhPC:POPA (red) or POPE:POPA (green) mixtures.
The following figure supplement is available for figure 2:

**Figure supplement 1**. Representative families of currents recorded from Kv channels.

In *Figure 3B* four lipids with different acyl chains but the same primary phosphate head-group are compared (*Figure 3—figure supplement 1*). These lipids are indistinguishable with respect to their effect on $V_{mid}$. Addition of either a methyl (DOPMe) or ethyl (DOPEth) group to the phosphate abolished the effect on gating (*Figure 3C*, *Figure 3—figure supplement 1*). On the other hand, addition of a second phosphate did not abolish the effect: the phosphodiester lipid DOPP appears to have a somewhat more potent effect on gating than DOPA (*Figure 3D*, *Figure 3—figure supplement 1*). When a free phosphate was present far away from the acyl chain, as in PIP, a $V_{mid}$ shift occurred but to a much lesser extent than in DOPA or DOPP. Thus, it would appear that $V_{mid}$ is mainly responsive to the presence of a primary phosphate group located relatively near the glycerol backbone.

## Asymmetric effect of POPA in the inner and outer membrane leaflets

In our experience Kv channels incorporate randomly into planar bilayers with approximately half the channels oriented outside-out (defined as extracellular surface of the channel facing the ground electrode) and half inside-out (defined as intracellular surface of the channel facing the ground electrode). If the membrane is held at −110 mV relative to ground (0 mV) and then stepped toward more positive voltages, only the outside-out channels open because the inside-out channels are inactivated. If the same membrane is held at +110 mV relative to ground and stepped toward more negative voltages, only the inside-out channels open because the outside-out channels are inactivated. *Figure 4A* shows the activation curves for outside-out and inside-out channels in the same membrane. The curves are indistinguishable. This is expected because in these experiments the lipid bilayer and ionic solutions on both sides of the membrane were symmetrical with respect to the inner and outer membrane leaflets.

When phospholipase D1 from *Streptomyces chromofuscus* was added to one side of a DPhPC membrane an asymmetry was generated because this enzyme cleaves the choline head-group and generates PA on the side to which it is added (*Ramu et al., 2006*). *Figure 4B* shows activation curves for outside-out channels recorded over time after addition of phospholipase D1 to the ground electrode side of the membrane (*Figure 4—figure supplement 1*). Because the ground electrode side of outside-out channels corresponds to the physiological extracellular surface of the channel, it is evident that the activation curve shifted to more negative voltages ($V_{mid}$ −65 to −87 mV) as PA was generated in the physiological extracellular leaflet (*Figure 4B*, *Figure 4—figure supplement 1*). *Figure 4C* shows activation curves for inside-out channels after phospholipase D1 was added to the ground electrode side of the membrane (*Figure 4—figure supplement 1*). Here we observe that as PA was generated

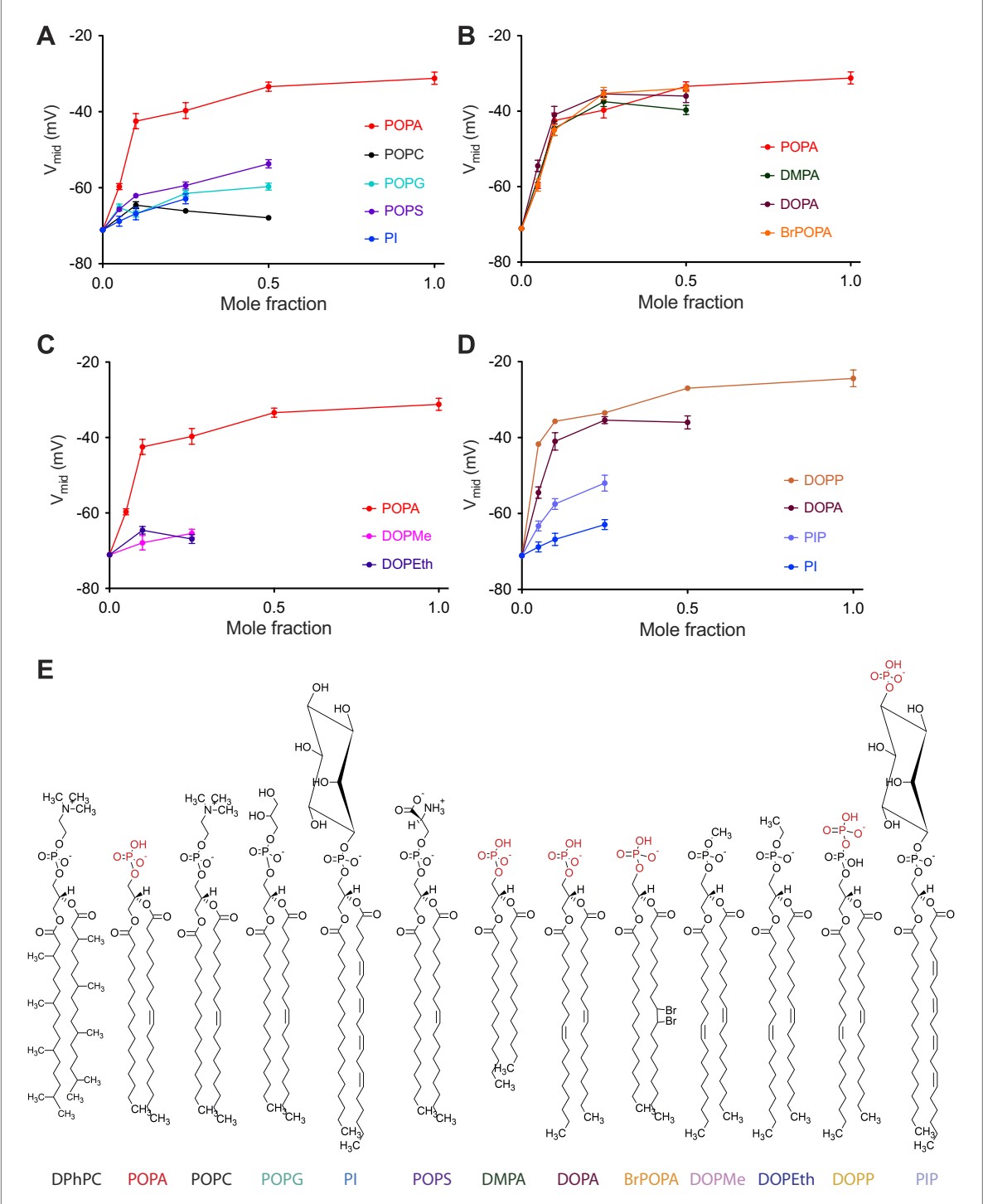

**Figure 3**. Concentration dependence of Kv channel activation by phospholipids. (**A**) Plot of $V_{mid}$ determined from a fit of tail currents to the Boltzmann equation vs phospholipid mole fraction for Kv channels in bilayers containing DPhPC:POPA (red), DPhPC:POPC (black), DPhPC:POPG (teal), DPhPC:POPS (purple) and DPhPC:PI (blue) mixtures. (**B**) Plot of $V_{mid}$ determined from a fit of tail currents to the Boltzmann equation vs phospholipid mole fraction for Kv channels in bilayers containing DPhPC:POPA (red), DPhPC:DMPA (green), DPhPC:DOPA (burgundy) and DPhPC:BrPOPA (orange) mixtures. (**C**) Plot of $V_{mid}$ determined from a fit of tail currents to the Boltzmann equation vs phospholipid mole fraction for Kv channels in bilayers containing DPhPC:POPA (red), DPhPC:DOPMe (pink) and DPhPC:DOPEth (blue) mixtures. (**D**) Plot of $V_{mid}$ determined from a fit of tail currents to the Boltzmann equation vs phospholipid mole fraction for Kv channels in bilayers containing DPhPC:DOPP (orange), DPhPC:DOPA (burgundy), DPhPC:PIP (violet) and DPhPC:PI (blue) mixtures. (**E**) Molecular structures of phospholipids analyzed in **A**–**D** with primary phosphates highlighted in red.

*Figure 3. Continued on next page*

*Figure 3. Continued*

The following figure supplement is available for figure 3:

**Figure supplement 1**. Representative families of currents recorded from Kv channels.

on the intracellular side of the channels the activation curve shifted toward more positive voltages. The $V_{mid}$ shift toward more positive voltages (~+40 mV) was greater than the shift toward more negative voltages (~−15 mV) (*Figure 4D*). PA thus has an opposite and greater effect on channel gating when acting on the intracellular membrane leaflet.

The PA composition of individual inner and outer membrane leaflets were altered another way. After formation of a DPhPC bilayer POPA was added in the form of vesicles to one side of the membrane. *Figure 5A* shows an activation curve for channels before and after addition of POPA to the ground electrode side. The effect of the POPA on outside-out and inside-out channels is shown. Here, as in the experiment in which PA was generated by enzymatic cleavage, shifts in the midpoint voltage of the activation curve were observed. POPA on the extracellular side (outside-out channels) produced a modest negative $V_{mid}$ shift (~−20 mV) while POPA on the intracellular side (inside-out channels) produced a large positive $V_{mid}$ shift (~+60 mV) (*Figure 5E*). These data are explicable if fused vesicles add their lipids predominantly to one leaflet. Accordingly, addition of POPA vesicles to the extracellular and intracellular leaflets had the same effect as adding phospholipase D1 to the extracellular and intracellular sides, respectively. Notably, both methods of PA addition had a greater effect on $V_{mid}$ when PA was altered on the intracellular leaflet.

## Characteristics of a surface charge voltage offset

Addition of electrically net neutral phospholipid POPC vesicles to either side of the membrane had little effect on $V_{mid}$ (*Figure 5B*). In contrast, addition of the −1 charged phospholipid POPG produced an approximately −20 mV shift when added to the extracellular side of the membrane and a +20 mV shift when added to the intracellular side (*Figure 5C*). A similar effect was observed when the −1 charged phospholipid POPS was used in place of POPG (*Figure 5D*). The near absence of an effect of neutral POPC and approximately equal positive and negative shifts in response to asymmetrically applied −1 charged POPG and POPS to the intracellular and extracellular surfaces, respectively, is consistent with a surface charge voltage offset.

A pictorial description of a surface charge voltage offset is shown in *Figure 6A–C*. Voltage sensors of Kv channels respond to the electric field inside the membrane, which is a function of both the applied voltage across the membrane and the surface potentials at the membrane–water interface. (The electric field is also a function of dipole potentials at the membrane water interface, but these, being essentially identical but oppositely oriented on the two sides of the membrane, cancel.) To interpret *Figure 6A–C*, consider that the channel responds to $V_{mem}$, the value of the voltage difference across the membrane where the voltage sensors reside. We the experimenters do not know the value of $V_{mem}$, but set the command voltage, $V_i − V_o$, with our amplifier. The pictures illustrate how changing surface charge on the membrane gives rise to different values of $V_{mem}$ under a constant command voltage according to the expression $V_{mem} = (V_i − V_o) + (\Phi_i − \Phi_o)$, where $\Phi_i$ is the surface potential on the inside and $\Phi_o$ is the surface potential on the outside. Now consider a channel whose half open probability occurs at a particular value, $V_{mem}'$. In the absence of surface charge the experimenter observes the half open probability at a command voltage $(V_i − V_o)_{mid} = V_{mem}'$. Upon addition of surface charge to the inner or outer leaflets the experimenter observes the half open probability at command voltage $(V_i − V_o)_{mid} = V_{mem}' + (\Phi_o − \Phi_i)$. Thus, negative surface charge on the outside shifts $V_{mid}$ towards more negative values and negative surface charge on the inside shifts $V_{mid}$ towards more positive values.

The relationship between membrane surface potential $\Phi$ (mV) and charge density $\sigma$ (electron charges per Å²) in a monovalent electrolyte (e.g. KCl) at concentration c (M) is

$$\Phi = \frac{2K_B T}{e_0} \sinh^{-1}\left(\frac{136\sigma}{\sqrt{c}}\right) \tag{1}$$

where $K_B$ is Boltzmann's constant, T is absolute temperature and $e_o$ the charge of an electron (*McLaughlin et al., 1970*). Using this expression and the mean surface area of a DPhPC molecule

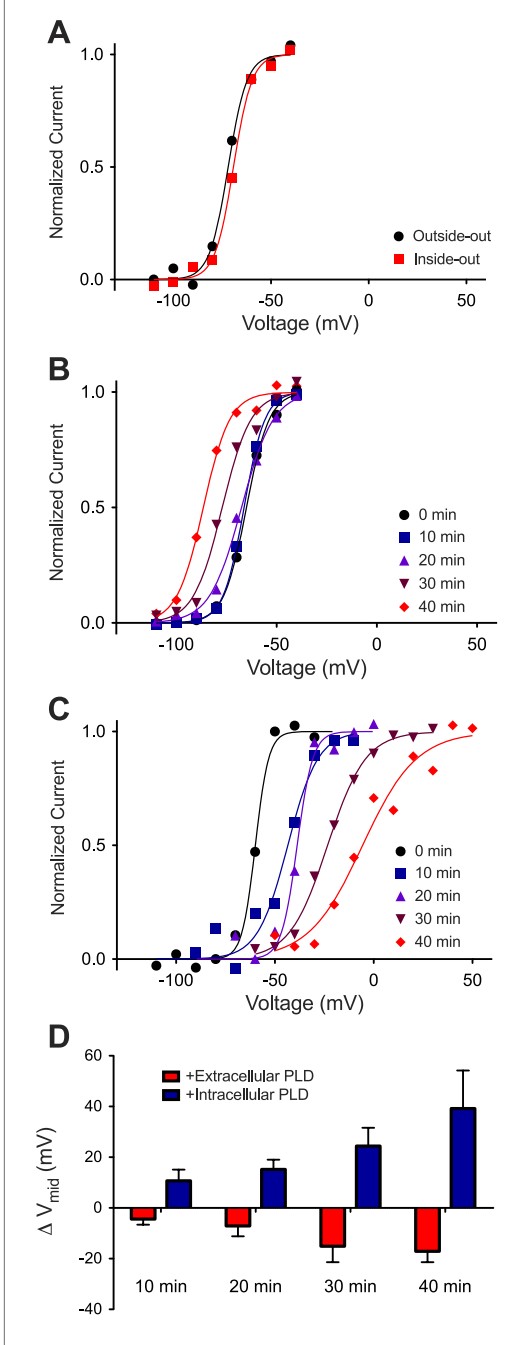

**Figure 4**. Kv activation in Phospholipase D1-treated DPhPC bilayers. (**A**) Normalized tail currents from representative current families recorded from Kv channels in a DPhPC bilayer (black–outside-out facing channels, $V_{mid}$ = −72 mV; red–inside-out facing channels, $V_{mid}$ = −69 mV) are fit to the Boltzmann equation. (**B**) Normalized tail currents from representative current families recorded from Kv channels in a DPhPC bilayer following addition of 50 units/ml *S. chromofuscus* phospholipase D1 to the extracellular membrane are fit to the Boltzmann equation (black 0 min, $V_{mid}$ = −65 mV; blue 10 min, $V_{mid}$ = −66 mV; *Figure 4. Continued on next page*

(~80 Å$^2$) (*Tristram-Nagle et al., 2010*) the 20 mV shifts in *Figure 5C,D* point to a surface charge density of about 1 $e_o$/900 Å$^2$ (neglecting contributions due to decane in the bilayer), which corresponds to a ratio of 1 POPG (or POPS) molecule per 11 DPhPC molecules, or a mole fraction of about 0.09. The equal magnitude but opposite direction shift produced by extracellular and intracellular POPG (or POPS) is consistent with a pure surface charge voltage offset (*Figure 5C,D*). The shift produced by extracellular POPA—similar in direction and magnitude to the shifts produced by extracellular POPG and POPS—is also consistent with a surface charge voltage offset. In contrast, the larger shift produced by intracellular POPA implies that an additional 'PA-specific' voltage offset is occurring (*Figure 5E*).

## Magnitude and origins of the PA-specific voltage offset

In the experiments shown in *Figures 2 and 3* both membrane leaflets contained the same lipid composition and therefore surface charge effects on the voltage sensor should have largely canceled. Therefore the $V_{mid}$ shift in these experiments must have resulted from a non-surface charge, PA-specific effect. In the experiments behind *Figures 4 and 5* the lipid composition of the two leaflets was different and therefore different surface charge densities should have contributed to the $V_{mid}$ shift. As noted earlier, POPA in the inner leaflet caused a greater shift than POPA in the outer leaflet or than POPG/POPS in either leaflet (*Figures 4B–D and 5A–E*). This can be understood in terms of a specific offset added to a surface charge offset for the case of POPA in the inner leaflet. In fact the shift produced by inner leaflet POPA (~+50 mV, *Figures 4C and 5A*) was approximately equal to the surface charge shift (~+20 mV, *Figure 5C,D*) plus the PA-specific shift (~+30 mV, *Figure 3A*). Therefore POPA, like other anionic lipids, imposes a bias on the voltage sensor because it creates a layer of negative charge on the membrane surface. But unlike other lipids POPA also exerts an additional bias from the inner membrane leaflet.

The surface charge effect is a simple electrostatic consequence of a fixed charge layer on the membrane surface. What is the origin of the PA-specific effect? A strong interaction between guanidinium and phosphate groups has been described (*Woods and Ferre, 2005*). We therefore wondered whether PA in the inner membrane leaflet might stabilize a closed conformation of the voltage sensor through interactions with one or more of the arginine residues on the

*Figure 4. Continued*

purple 20 min, $V_{mid} = -68$ mV; burgundy 30 min $V_{mid} = -77$ mV; red 40 min, $V_{mid} = -87$ mV). (**C**) Normalized tail currents from representative current families recorded from Kv channels in a DPhPC bilayer following addition of 50 units/ml *S. chromofuscus* phospholipase D1 to the intracellular side of the membrane are fit to the Boltzmann equation (black 0 min, $V_{mid} = -60$ mV; blue 10 min, $V_{mid} = -43$ mV; purple 20 min, $V_{mid} = -38$ mV; burgundy 30 min, $V_{mid} = -23$ mV; red 40 min, $V_{mid} = -6$ mV). (**D**) Average change in $V_{mid}$ following addition of 50 units/ml *S. chromofuscus* phospholipase D1 to the intracellular or extracellular side of the membrane ($\Delta V_{mid} = V_{mid}$ (t)–$V_{mid}$ (t = 0); 10 min intracellular $\Delta V_{mid} = 11$ mV, extracellular $\Delta V_{mid} = -4$ mV, N = 5; 20 min intracellular $\Delta V_{mid} = 15$ mV, extracellular $\Delta V_{mid} = -7$ mV, N = 5; 30 min intracellular $\Delta V_{mid} = 24$ mV, extracellular $\Delta V_{mid} = -15$ mV, N = 4; 40 min intracellular $\Delta V_{mid} = 39$ mV, extracellular $\Delta V_{mid} = -17$ mV, N = 2).

The following figure supplement is available for figure 4:

**Figure supplement 1**. Representative families of currents recorded from Kv channels in Phospholipase D1-treated DPhPC bilayers.

charge-bearing S4 helix of the voltage sensor. To test this possibility we turned to mutagenesis. Mutations in the paddle chimera Kv channel are not well tolerated so we tested whether the PA-specific effect is present in another distantly related Kv channel that can be more easily mutated. *Figure 7A* shows a PA-specific (both membrane leaflets contain PA so the surface charge effect should be absent) positive $V_{mid}$ shift in the activation curve of the archeal channel KvAP (*Figure 7—figure supplement 1*). The shift is smaller (+17 mV) than that observed in the eukaryotic Kv channel (+30 mV) but clearly present (*Figures 2A and 7A*). *Figure 7B* graphs the $V_{mid}$ shift brought about by DPhPA in wild type KvAP and in several mutants within S4 (*Figure 7—figure supplement 1*). Replacement of the arginine at position 133 by either lysine or alanine abolished the DPhPA-induced shift. Introduction of an arginine at position 136 restored the shift in the absence of an arginine at 133.

A plausible interpretation is that the S4 arginine at position 133—and at position 136 in the context of no arginine at position 133—can interact with a primary phosphate group in the membrane's inner leaflet. *Figure 8A* shows x-ray crystal structures of the paddle chimera and KvAP voltage sensors (*Jiang et al., 2003*; *Long et al., 2007*). Arginine 133 in KvAP and the corresponding arginine in paddle chimera are located close to the center of the membrane bilayer (*Figure 8A*). But these structures correspond to 'open' conformations of voltage sensors. In closed conformations the centrally located arginine residues likely approach the membrane's inner leaflet where they could interact with PA to help stabilize that conformation.

## Discussion

Phosphatidic acid is present in many cellular membranes including the inner leaflet of the plasma membrane where it plays essential roles in cellular pathways such as mTOR complex stability and signaling (reviewed in *Foster, 2013*), growth factor receptor signaling (reviewed in *Gomez-Cambronero, 2010*) and hormone signaling (*Garrido et al., 2009*). In its signaling role, PA is most commonly generated from either phosphatidyl choline via phospholipase D cleavage or from diacylglycerol via diacylglycerol kinase (*Foster, 2013*). Once generated, PA can then be rapidly depleted from the plasma membrane by phosphatidic acid phosphatase or a variety of phospholipases, thus allowing tight cellular control over its abundance (*Foster, 2013*). The precise control of plasma membrane PA concentrations in response to intracellular and extracellular stimuli combined with its influence on Kv channel function situate PA at a potential interface between cellular and global metabolic signaling pathways and membrane excitation.

Given the importance of PA to cellular signaling we suspect that the unique ability of PA to alter Kv channel gating is of biological significance. PA's effect is exerted by two mechanisms. The surface charge component is expected of any charged lipid disposed asymmetrically over the two membrane leaflets (*Ramu et al., 2006*; *Xu et al., 2008*). The PA-specific component requires a primary phosphate group on the lipid molecule and acts only from the inner leaflet. The mutational data support a hypothesis that PA in the inner leaflet interacts with specific arginine residues in the voltage sensor. It seems plausible that hydrogen bonding between a primary phosphate group and arginine guanidinium group could hold the voltage sensor closed, requiring stronger depolarization to open (i.e. a $V_{mid}$ shift to more positive voltages). We have determined crystal structures of the paddle chimera mutant in the presence of brominated PA. We typically observe electron density for lipid molecules in the crystal structure, but none with a specific signal for bromine. Therefore we do not know whether PA is

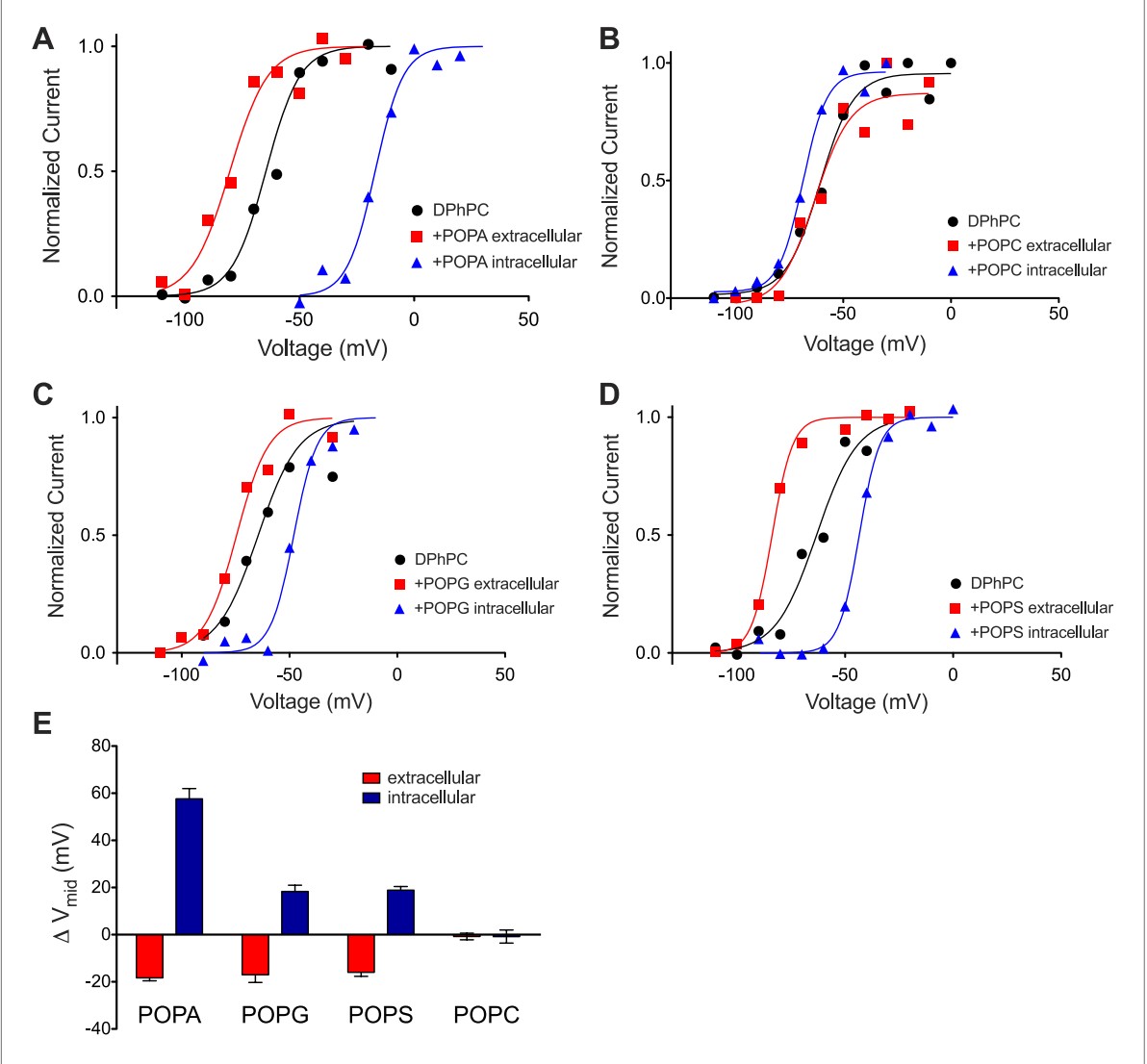

**Figure 5**. Kv activation in DPhPC bilayers fused with phospholipid vesicles. (**A**) Normalized tail currents from representative current families recorded from Kv channels in a DPhPC bilayer before (black, $V_{mid}$ = −65 mV) and after fusion of POPA vesicles to the extracellular (red, $V_{mid}$ = −80 mV) or intracellular (blue, $V_{mid}$ = −17 mV) surface of the bilayer are fit to the Boltzmann equation. (**B**) Normalized tail currents from representative current families recorded from Kv channels in a DPhPC bilayer before (black, $V_{mid}$ = −61 mV) and following fusion of POPC vesicles to the extracellular (red, $V_{mid}$ = −63 mV) or intracellular (blue, $V_{mid}$ = −69 mV) surface of the bilayer are fit to the Boltzmann equation. (**C**) Normalized tail currents from representative current families recorded from Kv channels in a DPhPC bilayer before (black, $V_{mid}$ = −65 mV) and following fusion of POPG vesicles to the extracellular (red, $V_{mid}$ = −76 mV) or intracellular (blue, $V_{mid}$ = −48 mV) surface of the bilayer are fit to the Boltzmann equation. (**D**) Normalized tail currents from representative current families recorded from Kv channels in a DPhPC bilayer before (black, $V_{mid}$ = −63 mV) and following fusion of POPS vesicles to the extracellular (red, $V_{mid}$ = −83 mV) or intracellular (blue, $V_{mid}$ = −43 mV) surface of the bilayer are fit to the Boltzmann equation. (**E**) Average change in $V_{mid}$ following addition of phospholipid vesicles to the intracellular or extracellular side of the membrane ($\Delta V_{mid}$ = $V_{mid}$ (vesicle fusion)–$V_{mid}$ (no fusion); POPA extracellular $\Delta V_{mid}$ = −18 mV, intracellular $\Delta V_{mid}$ = 58 mV, N = 4; POPG extracellular $\Delta V_{mid}$ = −17 mV, intracellular $\Delta V_{mid}$ = −18 mV, N = 4; POPS extracellular $\Delta V_{mid}$ = −16 mV, intracellular $\Delta V_{mid}$ = 19 mV, N = 4; POPC extracellular $\Delta V_{mid}$ = −1 mV, intracellular $\Delta V_{mid}$ = −1 mV, N = 3).

specifically bound to a groove on the surface of the channel. Given the somewhat surprising observation that PA influences gating similarly in both the eukaryotic Kv channel and KvAP—channels that are only distantly related—we imagine that PA's specific effect might be mediated through the guanidinium–phosphate interaction alone (i.e. without other interactions between the lipid tail and the channel). In other words, we imagine in a closed conformation the arginine could be released from its hydrogen bond pairing with counter charges on the channel and become anchored in the membrane's inner leaflet through hydrogen bonding with a primary phosphate group.

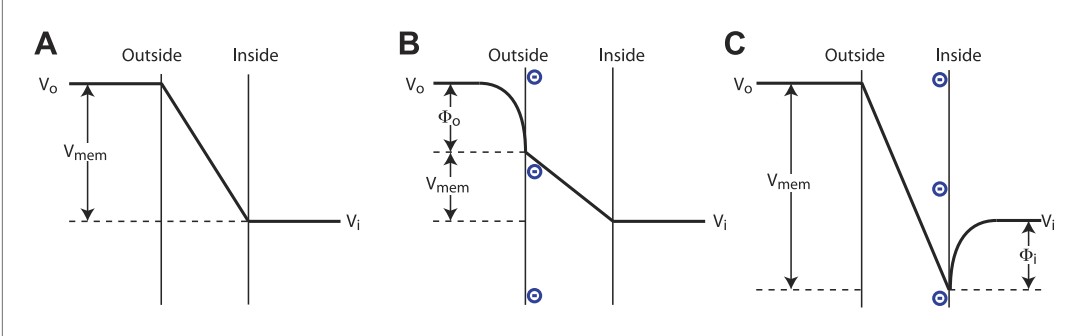

**Figure 6**. Surface charge voltage offset in phospholipid membranes. (**A**) In symmetric membranes lacking charged phospholipids, $V_{mem}$, the voltage to which channels respond, is equal to the command voltage, $V_i - V_o$, set on the amplifier. (**B**) In asymmetric membranes containing anionic lipids exclusively in the outer leaflet of the membrane, $V_{mem}$ is equal to the command voltage, $V_i - V_o$, minus the surface potential of the outer membrane, $\Phi_o$. (**C**) In asymmetric membranes containing anionic lipids exclusively in the inner leaflet of the membrane, $V_{mem}$ is equal to the command voltage, $V_i - V_o$, plus the surface potential of the inner membrane, $\Phi_i$.

The magnitude of the voltage shift is definitely large enough to influence the electrical properties of an excitable cell. A shift towards more positive voltages brought about by increasing PA in the inner leaflet will 'silence' $K^+$ channels over an otherwise active voltage range. This silencing will lead to increased membrane excitability. The sensitivity of Kv channels to PA seems a likely link between metabolic pathways coupled to lipid metabolism and membrane excitability.

## Materials and methods

### Kv channel purification and reconstitution

A mutant of the rat Kv1.2 channel in which the helix-turn-helix segment termed the voltage sensor paddle was replaced by the corresponding segment from Kv2.1, known as the paddle chimera Kv channel, was expressed and purified as described previously (***Long et al., 2007***; ***Tao and MacKinnon, 2008***), with minor modifications. In brief, the paddle chimera Kv channel was co-expressed with the rat β2-core gene in *Pichia pastoris*. The channel complex was extracted from membranes with DDM (Anatrace, Maumee, OH) and purified with a cobalt affinity column followed by size exclusion chromatography on a Superdex-200 gel filtration column (GE Biosciences, Pittsburgh, PA). The size exclusion buffer was composed of 20 mM Tris–HCl, pH 7.5, 150 mM KCl, 6 mM DM (Anatrace), 2 mM Tris(2-carboxyethyl)phosphine, 2 mM dithiothreitol, and 0.1 mg/ml POPC:POPE:POPG 3:1:1 (mass ratio) (Avanti Polar Lipids, Alabaster, AL).

Purified channel complexes were reconstituted into OM (Anatrace)-solubilized 3:1 (wt:wt) POPE:POPG lipid vesicles as described (***Long et al., 2007***). Detergent was removed by dialysis for 5 days against detergent-free buffer containing 10 mM HEPES-KOH, pH 7.5, and 450 mM KCl, and 2 mM dithiothreitol at 4°C, with daily buffer exchanges. After 5 days, all residual detergent was removed by incubating the reconstituted channels with Bio-Beads (Bio-Rad, Hercules, CA) for 2 hr at room temperature. The reconstituted channels were aliquoted and flash frozen into liquid nitrogen prior to storage at −80°C.

### KvAP purification and reconstitution

KvAP was expressed and purified as described previously (***Ruta et al., 2003***). Briefly, the channel was extracted from *Escherichia coli* membranes with DM (Anatrace) and purified with a cobalt affinity column followed by size exclusion chromatography on a Superdex-200 gel filtration column (GE Biosciences) with 20 mM Tris–HCl, pH 8.0, 100 mM KCl, 4 mM DM (Anatrace). Purified KvAP channels were reconstituted into DM-solubilized 3:1 (wt:wt) POPE:POPG lipid vesicles at a 1:10 (wt:wt) protein:lipid ratio. Detergent was removed by dialysis for 3 days against detergent-free buffer containing 10 mM HEPES, 4 mM N-methylglucamine, pH 7.4, and 450 mM KCl, with twice daily buffer exchanges. The reconstituted channels were aliquoted and flash frozen into liquid nitrogen prior to storage at −80°C.

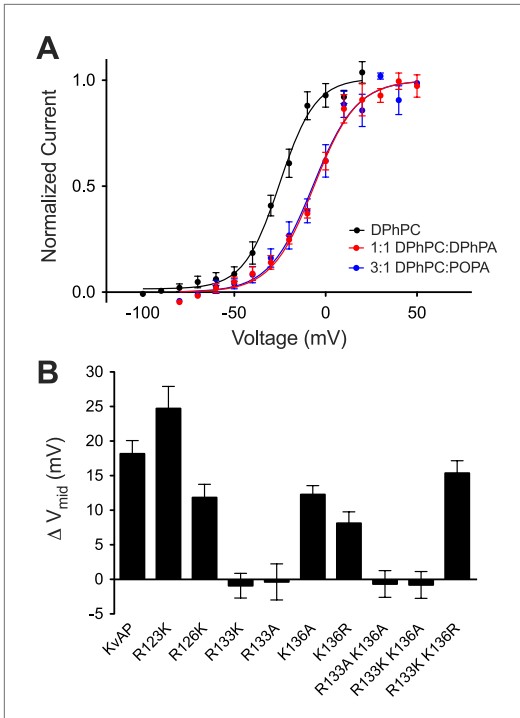

**Figure 7**. Phosphatidic acid modifies KvAP activation. (**A**) Normalized tail currents (mean ± SEM) from current families recorded from KvAP in DPhPC bilayers (black, $V_{mid}$ = −25 ± 1 mV, Z = 2.6, N = 8), DPhPC:DPhPA (1:1) bilayers (red, $V_{mid}$ = −6 ± 1 mV, Z = 2.2, N = 6) and DPhPC:POPA (3:1) bilayers (blue, $V_{mid}$ = −7 ± 2 mV, Z = 2.2, N = 5) are fit to the Boltzmann equation. (**B**) Average difference in $V_{mid}$ between DPhPC and DPhPC:DPhPA (1:1) membranes for KvAP and KvAP mutant channels ($\Delta V_{mid}$ = $V_{mid}$ (DPhPC:DPhPA 1:1)–$V_{mid}$ (DPhPC)). The bar heights correspond to KvAP 18 mV, N = 5; R123K 24 mV, N = 5; R126K 12 mV, N = 5; R133K −1 mV, N = 6; R133A 0 mV, N = 5; K136A 12 mV, N = 5; K136R 8 mV, N = 5; R133A K136A −1 mV, N = 6; R133K K136A −1 mV, N = 5; R133K K136R 15 mV, N = 5.

The following figure supplement is available for figure 7:

**Figure supplement 1**. Representative families of currents recorded from KvAP channels.

## Electrophysiological recordings from planar lipid bilayers

Planar lipid bilayer experiments were performed as described previously (*Miller, 1986*; *Ruta et al., 2003*). Lipids of desired compositions were prepared by dissolving argon-dried lipids in decane to a final concentration of 20 mg/ml. Lipid solutions were painted over a 300 µm hole in a polystyrene partition that separated the two chambers to form the planar lipid bilayer. The chamber (*cis*) contained 4 ml of 150 mM KCl and 10 mM HEPES-KOH, pH 7.5, while the cup (*trans*) contained 3 ml of 15 mM KCl and 10 mM HEPES-KOH, pH 7.5. Reconstituted channels were pipetted onto the chamber side of the bilayer after thinning of a planar lipid bilayer had been detected via monitoring of electrical capacitance. Once channels were successfully fused with the bilayer, 135 mM KCl was added to the cup side to equilibrate the $K^+$ concentrations across the bilayer. Membranes were held at a negative holding voltage, stepped to more depolarized voltages in 10-mV increments and then back to the negative holding voltage to close the channels. Shortly after the return to the negative holding voltage, inward current called 'tail current' is measured. The fraction of maximal activation at each depolarization voltage can be determined by graphing the inward tail current, normalized by the maximum value, as a function of the preceding depolarization voltage and fit to a two-state Boltzmann equation:

$$\frac{I}{I_{max}} = \frac{1}{1+e^{\frac{-ZF}{RT}(V-V_{mid})}} \qquad (2)$$

where $I/I_{max}$ is the fraction of maximal current, V is the command depolarization voltage to open the channels, $V_{mid}$ is the command voltage at which the channels have reached 50% of their maximal current, F is the Faraday constant, R is the gas constant, T is the absolute temperature and Z is the apparent valence of the voltage dependence.

All recordings were performed using the voltage-clamp method in whole-cell mode. Analogue signals were filtered at 1 kHz using a low-pass Bessel filter on an Axopatch 200B amplifier (Molecular Devices) in whole-cell mode and digitized at 10 kHz using a Digidata 1400A analogue-to-digital converter (Molecular Devices). The pClamp software suite (Molecular Devices) was used to control membrane voltage and record current.

## Phospholipase D1 generation of phosphatidic acid

Following successful incorporation of paddle chimera channels into a DPhPC bilayer, 4 µl of 50,000 units/ml phospholipase D1 purified from *S. chromofuscus* (Sigma, St. Louis, MO) were pipetted into the ground side of the bilayer and mixed thoroughly with a pipette, resulting in a final concentration of 50 units/ml. Electrophysiological recordings were conducted using both the forward and reverse protocols every ten minutes until the membrane became unstable.

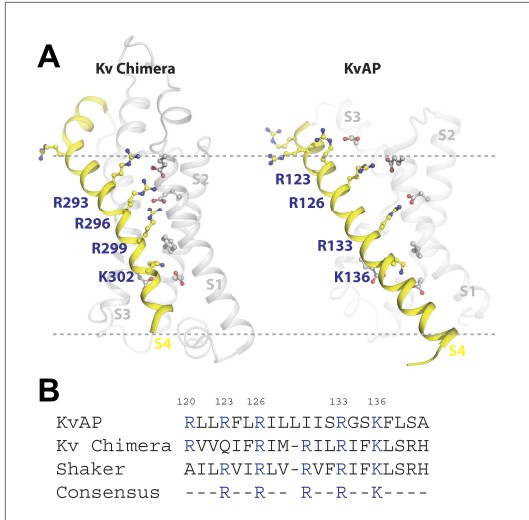

**Figure 8**. Structure and sequence alignment of Kv voltage sensor domains. (**A**) View from the membrane plane of the voltage sensor domains of rat Kv1.2–2.1 paddle chimera and KvAP. The S4 transmembrane helices and their positively charged residues are highlighted in yellow. The dashed grey line marks the approximate positions of the phospholipid head groups. (**B**) Sequence alignment of the S4 transmembrane helices from rat Kv1.2–2.1 paddle chimera, KvAP and *D. melanogaster* Shaker K+ channel. Numbering is according to KvAP sequence.

## Vesicle fusion

10 mg of desired lipid in chloroform were dried down under constant Argon stream and then resuspended into 0.5 ml of 450 mM KCl, 10 mM HEPES pH 7.5 by sonication, resulting in a final concentration of 20 mg/ml. Following incorporation of paddle chimera channels into a DPhPC bilayer, 1.5 µl of the vesicle solution was pipetted onto the chamber side of the bilayer. After 10 min, electrophysiological recordings were performed using both the forward and reverse protocols from the same bilayer.

## Abbreviations

DPhPC–1,2-diphytanoyl-*sn*-glycero-3-phosphocholine, POPA–1-palmitoyl-2-oleoyl-*sn*-glycero-3-phosphate, POPC–1-palmitoyl-*sn*-glycero-3-phosphocholine, POPG 1–palmitoyl-2-oleoyl-*sn*-glycero-3-phosphoglycerol, PI–Bovine liver L-α-phosphatidylinositol, POPS–1-palmitoyl-2-oleoyl-*sn*-glycero-3-phospho-L-serine, DMPA–1,2-dimyristoyl-*sn*-glycero-3-phosphate, DOPA–1,2-dioleoyl-*sn*-glycero-3-phosphate, BrPOPA–1–palmitoyl-2-(9,10-dibromo) stearoyl-*sn*-glycero-3-phosphate, DOPMe–1,2-dioleoyl-*sn*-glycero-3-phosphomethanol, DOPEth–1,2-dioleoyl-*sn*-glycero-3-phosphoethanol, DOPP–1,2-dioleoylglycerol pyrophosphate, PIP–Porcine brain L-α-phosphatidylinositol-4-phosphate, PIP$_2$-phosphatidylinositol-4,5-bisphosphate, Cardiolipin–Bovine heart Cardiolipin, Sphingomyelin–Porcine brain Sphingomyelin, DPhPA–1,2-diphytanoyl-*sn*-glycero-3-phosphate, OM_octyl-β-D-maltopyranoside, DDM–*n*-dodecyl-β-*D*-maltopyranoside, DM–*n*-decyl-β-*D*-maltopyranoside, HEPES–N-(hydroxyethyl)piperazine-N'-2-ethanesulfonic acid.

## Acknowledgements

We thank Xiao Tao and Anirban Banerjee for advice on biochemistry and channel reconstitution and members of the MacKinnon laboratory for helpful discussions. This work was supported in part by GM43949. RKH is a Howard Hughes Medical Institute Fellow of The Helen Hay Whitney Foundation. RM is an investigator of the Howard Hughes Medical Institute.

## Additional information

### Funding

| Funder | Grant reference number | Author |
|---|---|---|
| Howard Hughes Medical Institute | | Roderick MacKinnon |
| National Institute of General Medical Sciences | GM43949 | Richard K Hite, Joel A Butterwick, Roderick MacKinnon |

The funders had no role in study design, data collection and interpretation, or the decision to submit the work for publication.

### Author contributions

RKH, Conception and design, Acquisition of data, Analysis and interpretation of data, Drafting or revising the article; JAB, Acquisition of data, Analysis and interpretation of data, Drafting or revising

the article, Contributed unpublished essential data or reagents; RM, Conception and design, Analysis and interpretation of data, Drafting or revising the article

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
