## [Decision Letter]

Thank you for sending your work entitled “Phosphatidic acid modulation of Kv channel voltage sensor function” for consideration at *eLife*. Your article has been favorably evaluated by John Kuriyan (Senior editor), Richard Aldrich (Reviewing editor) and two other reviewers, both of whom, Kenton Swartz and Chris Miller, have agreed to reveal their identities.

The Reviewing editor and the other reviewers discussed their comments before we reached this decision, and the Reviewing editor has assembled the following comments to help you prepare a revised submission.

This is an interesting manuscript describing a thoughtful investigation of the lipid dependence of gating of a mammalian voltage-activated potassium (Kv1) channel. The main channel studied here is the Kv1.2/2.1 paddle chimera, for which an X-ray structure is available, and the authors incorporate the channel into planar lipid membranes, using DPhPC as the standard control lipid. When comparing a wide range of additional lipids incorporated into both leaflets of planar lipid bilayers, the authors find that the phosphatidic acid (PA) DOPA produces the most dramatic effect, shifting the G-V relationship to more positive voltages. Further comparison of lipids with alterations in either the headgroups or acyl chains shows that this effect is due to the presence of an anionic headgroup. In elegant experiments with a phospholipase D, which will produce PA from DPhPC, the authors find different effects when modifying the external and internal leaflets. In the external leaflet, PA shifts the G-V to negative voltages whereas in the internal leaflet PA produces a much larger shift to more positive voltages. The authors then compare a series of anionic lipids (DOPA, POPG and POPS) with zwitterionic POPC in experiments in which vesicles containing those lipids are fused with planar lipid membranes containing DPhPC. All three anionic lipids produce negative shifts in the G-V when applied externally, but when applied internally, POPA produces a larger positive shift when compared to POPG or POPS. From these results the authors conclude that anionic lipids in external and internal leaflets shift the G-V relationships by altering the surface charge of the membrane, and that intracellular PA has an additional effect that is not as robust as the larger anionic lipids tested. In a final series of experiments with KvAP, another Kv channel for which structures are available and that is easier to mutate, express and reconstitute, the authors contend that this additional effect of internal PA can be explained by an interaction of the anionic lipid with the innermost basic residue positions of the S4 helix that stabilizes a closed state.

Overall, the reviewers really like the work, and think that careful revision of the manuscript could make an excellent paper that would be very appropriate for *eLife.*

Suggestions for revision:

1) While the paper is generally clear, the writing could be improved. Many sections of the Results begin by immediately presenting results without defining the questions for which answers are being sought or motivating the experiments that were done. At times awkwardly casual phrases are used.

2) The experimental data in this manuscript are macroscopic voltage-activation relationships determined by measuring tail currents following depolarizations to a range of different membrane voltages, and these are fit with single Boltzmann functions to obtain Vmid. We appreciate that the authors are breaking new ground, and are trying to not get bogged down, but it would be helpful to the reader to have a bit more information than simply G-V relations and Vmid values. For most of the experiments it would be nice to provide example traces so the reader can appreciate whether the various manipulations have interesting effects on the kinetics of activation, inactivation or deactivation, and it would be helpful to report the slopes of the fits of Boltzmann functions to the data. From the two families of current traces shown in Figure 1 its clear that interesting things are happening the kinetics of gating, and it would valuable to provide this type of information for as many experiments as possible.

3) For the experimental results presented in Figures 4 and 5, where manipulations were performed during the course of doing the experiment, it would be helpful to see whether the maximal macroscopic conductance changes. Does adding PLC to external or internal leaflets do more than shift Vmid?

4) It would be nice to provide a population measure for the experiments illustrated in Figure 4, perhaps as was done in Figure 5.

5) For all of the vesicle experiments, is there any way to know that vesicles are actually fusing? We would imagine that zwitterionic vesicles might not fuse nearly as efficiently as negatively charged vesicles.

6) The presentation of information depicted in Figure 6 needs work and will likely make readers more confused rather than less. There are many inconsistencies between the text, figure and caption. The authors need to go through this very carefully and do a much better job of explaining, defining and labeling things.

Discussion of Figure 6 defines PHI as applied voltage, but in Figure 6 PHI is potential difference across the membrane.

In parts A, B and C it would help to clarify that the dark line is electric potential.

The voltage markings shown appear to be voltage difference between +/- infinity. Thus V is the voltage applied by the experimenter, but the caption claims that V is the “mean value of the electric field”.

PHI: it is unclear why the lower dashed line extends into intracellular solution (suggest shortening to stop at the inner membrane surface). Its unclear why the upper dashed line extends across membrane (shorten to stop at outer membrane surface)

As positioned/marked, PHI is the potential difference between the inner and outer surfaces of the membrane, yet the caption claims PHI is “the applied voltage”.

\Psi_i, \Psi_o “i” index is on outer surface -> should be “o” and “o” index is on inner surface -> should be “i”

In Figure 6 it would be easier to follow if Vmid was used instead of |deltaVmid|, which isn't even properly define. Maybe a bar plot like Figure 5 with Vmid for:

POPA ext

POPA int

POPA sym

POPG Int

POPA sym + POPG/PS Int

7) It would be helpful to have more justification and information on the mutagenesis results presented in Figure 7. Is KvaP really a valid surrogate for the paddle chimera in these experiments? From reading the early Schmidt papers it would appear that KvAP in membranes containing the anionic POPG have G-Vs shifted to negative membrane voltages and when containing positively charged Eth-DOPC they are shifted positive. Are these responses to lipid composition comparable to what is seen with paddle chimera? Also, it would be important to present both the Vmid values along with deltaVmid for the S4 mutations. If the mutations alone have large effects on gating transitions then its possible the apparent effects of PA would appear less.

8) There is a body of work showing that PUFAs modulate the Shaker Kv channel through an electrostatic mechanism and it would be appropriate to cite and refer to that work in this manuscript.

---

## [Author Response]

*1) While the paper is generally clear, the writing could be improved. Many sections of the Results begin by immediately presenting results without defining the questions for which answers are being sought or motivating the experiments that were done. At times awkwardly casual phrases are used*.

We have modified the text to make it easier for readers to follow.

*2) The experimental data in this manuscript are macroscopic voltage-activation relationships determined by measuring tail currents following depolarizations to a range of different membrane voltages, and these are fit with single Boltzmann functions to obtain Vmid. We appreciate that the authors are breaking new ground, and are trying to not get bogged down, but it would be helpful to the reader to have a bit more information than simply G-V relations and Vmid values. For most of the experiments it would be nice to provide example traces so the reader can appreciate whether the various manipulations have interesting effects on the kinetics of activation, inactivation or deactivation, and it would be helpful to report the slopes of the fits of Boltzmann functions to the data. From the two families of current traces shown in*
Figure 1
*its clear that interesting things are happening the kinetics of gating, and it would valuable to provide this type of information for as many experiments as possible*.

The valences for the fits to the Boltzmann equation have been added to the figure legends. We have also added representative traces for each of the lipid mixtures examined in Figure 1 as Figure 1—figure supplement 1. Representative traces for each of the DPhPC:POPA ratios shown in Figure 2 have been added as Figure 2—figure supplement 1. Representative traces for each of the previously untested lipids in DPhPC at a 3:1 molar ratio have been added as Figure 3—figure supplement 1. Representative traces of KvAP in DPhPC, DPhPC:DPhPA (1:1) and DPhPC:POPA (3:1) have been added as Figure 7—figure supplement 1.

*3) For the experimental results presented in*
Figures 4 and 5*, where manipulations were performed during the course of doing the experiment, it would be helpful to see whether the maximal macroscopic conductance changes. Does adding PLC to external or internal leaflets do more than shift Vmid?*

The experiments presented in Figures 4 and 5 are technically challenging due to the need to maintain a stable membrane for an extended time course while pulsing from opposite holding voltages. In order to increase the likelihood of obtaining a successful experiment, we minimized the number of pulses per voltage family while still being able to fit a Boltzmann to the charge-conductance curve. Figure 4—figure supplement 1 shows representative traces of Kv channels 0 and 40 minutes after addition of PLD and they are qualitatively similar.

*4) It would be nice to provide a population measure for the experiments illustrated in*
Figure 4*, perhaps as was done in*
Figure 5.

We have added an graph plotting the mean change in Vmid for the 10, 20, 30 and 40 minutes time points following addition of PLD to intracellular and extracellular sides of the membrane as Figure 4.

*5) For all of the vesicle experiments, is there any way to know that vesicles are actually fusing? We would imagine that zwitterionic vesicles might not fuse nearly as efficiently as negatively charged vesicles*.

We do not have carried out experiments to verify that vesicle fusion occurred during our experiments. For the anionic phospholipids, fusion produced reproducible shifts in the Vmid. For POPC, repeated attempts at fusion yielded no change in the Vmid. The absence of a shift following fusion of POPC vesicles was interpreted as an asymmetric distribution of POPC in a DPhPC membrane having no influence upon Kv channel activation. Without independent verification, however, it is not possible to exclude the possibility that the POPC vesicles failed to fuse with the membranes. Such a possibility does not alter our overall interpretation of the results, as the clear difference in responses between PA and the other anionic phospholipids would still remain.

*6) The presentation of information depicted in*
Figure 6
*needs work and will likely make readers more confused rather than less. There are many inconsistencies between the text, figure and caption. The authors need to go through this very carefully and do a much better job of explaining, defining and labeling things*.

*Discussion of*
Figure 6
*defines PHI as applied voltage, but in*
Figure 6
*PHI is potential difference across the membrane*.

*In parts A, B and C it would help to clarify that the dark line is electric potential*.

*The voltage markings shown appear to be voltage difference between +/- infinity. Thus V is the voltage applied by the experimenter, but the caption claims that V is the “mean value of the electric field”*.

PHI: it is unclear why the lower dashed line extends into intracellular solution (suggest shortening to stop at the inner membrane surface). Its unclear why the upper dashed line extends across membrane (shorten to stop at outer membrane surface)

*As positioned/marked, PHI is the potential difference between the inner and outer surfaces of the membrane, yet the caption claims PHI is “the applied voltage”*.

\Psi_i, \Psi_o “i” index is on outer surface -> should be “o” and “o” index is on inner surface -> should be “i”

*In*
Figure 6
*it would be easier to follow if Vmid was used instead of |deltaVmid|, which isn't even properly define. Maybe a bar plot like*
Figure 5
*with Vmid for:*

POPA ext

POPA int

POPA sym

POPG Int

POPA sym + POPG/PS Int

We have modified Figure 6 and altered the labeling of panels A-C in both the text and figure legend such that all are consistent with the figure. We have removed Figure 6 due to the confusion that it caused and instead conveyed the same information in the text (the sections Characteristics of a surface charge voltage offset and Magnitude and origins of the PA-specific voltage offset have been modified for clarity).

*7) It would be helpful to have more justification and information on the mutagenesis results presented in*
Figure 7*. Is KvaP really a valid surrogate for the paddle chimera in these experiments? From reading the early Schmidt papers it would appear that KvAP in membranes containing the anionic POPG have G-Vs shifted to negative membrane voltages and when containing positively charged Eth-DOPC they are shifted positive. Are these responses to lipid composition comparable to what is seen with paddle chimera? Also, it would be important to present both the Vmid values along with deltaVmid for the S4 mutations. If the mutations alone have large effects on gating transitions then its possible the apparent effects of PA would appear less*.

KvAP is not presented as a surrogate for paddle chimera. We refer to it as distantly related. We were surprised to see that POPA shifts its activation curve in a similar manner. That it does seems consistent with our suggestion in the discussion that POPA stabilizes the closed state through arginine interactions with primary phosphates in the inner leaflet rather than through binding to well-defined groove on the channel surface. In most other respects KvAP and paddle chimera are not very similar.

*8) There is a body of work showing that PUFAs modulate the Shaker Kv channel through an electrostatic mechanism and it would be appropriate to cite and refer to that work in this manuscript*.

PUFAs are generally thought to inhibit Kv channels, but there is not a consensus for the mechanism of action. Decher and colleagues suggested that PUFAs serve as pore blockers by binding to a site in the inner cavity of Kv1.1 channels expressed in oocytes (Decher et al, *EMBO J.* 29, 2101-2113.). In contrast, Oliver and colleagues suggest that PUFAs inhibit Kv channels by inducing conformational changes in the selectivity filter (Oliver et al, *Science.* 304, 265-270.). Due to the lack of consensus in the field, we would prefer not to include the discussion of PUFAs.